# Translational control of nicotine-evoked synaptic potentiation in mice and neuronal responses in human smokers by eIF2α

Andon N Placzek[1,2†‡], David L Molfese[3,4†], Sanjeev Khatiwada[1,2,5†], Gonzalo Viana Di Prisco[1,2], Wei Huang[1,2], Carmela Sidrauski[6§], Krešimir Krnjević[7], Christopher L Amos[8], Russell Ray[1,2], John A Dani[9], Peter Walter[6], Ramiro Salas[1,3,4], Mauro Costa-Mattioli[1,2*]

[1]Department of Neuroscience, Baylor College of Medicine, Houston, United States; [2]Memory and Brain Research Center, Baylor College of Medicine, Houston, United States; [3]Menninger Department of Psychiatry and Behavioral Sciences, Baylor College of Medicine, Houston, United States; [4]Michael E. DeBakey Veteran Administration Medical Center, Houston, United States; [5]Verna and Marrs McLean Department of Biochemistry and Molecular Biology, Baylor College of Medicine, Houston, United States; [6]Department of Biochemistry and Biophysics, Howard Hughes Medical Institute, University of California, San Francisco, San Francisco, United States; [7]Department of Physiology, McGill University, Montreal, Canada; [8]Center for Genomic Medicine, Department of Community and Family Medicine, Geisel School of Medicine, Dartmouth College, Lebanon, United States; [9]Department of Neuroscience, Mahoney Institute for Neurosciences, Perelman School of Medicine, Philadelphia, United States

*For correspondence: costamat@bcm.edu

†These authors contributed equally to this work

Present address: ‡Division of Basic Medical Sciences, Mercer University School of Medicine, Macon, United States; §Calico LLC, South San Francisco, United States

Competing interests: The authors declared that no competing interests exist.

**Abstract** Adolescents are particularly vulnerable to nicotine, the principal addictive component driving tobacco smoking. In a companion study, we found that reduced activity of the translation initiation factor eIF2α underlies the hypersensitivity of adolescent mice to the effects of cocaine. Here we report that nicotine potentiates excitatory synaptic transmission in ventral tegmental area dopaminergic neurons more readily in adolescent mice compared to adults. Adult mice with genetic or pharmacological reduction in p-eIF2α-mediated translation are more susceptible to nicotine's synaptic effects, like adolescents. When we investigated the influence of allelic variability of the *Eif2s1* gene (encoding eIF2α) on reward-related neuronal responses in human smokers, we found that a single nucleotide polymorphism in the *Eif2s1* gene modulates mesolimbic neuronal reward responses in human smokers. These findings suggest that p-eIF2α regulates synaptic actions of nicotine in both mice and humans, and that reduced p-eIF2α may enhance susceptibility to nicotine (and other drugs of abuse) during adolescence.

## Introduction

Tobacco use is a major global health problem with enormous economic and social costs. It remains the leading cause of preventable death worldwide, with tobacco-related illnesses estimated to kill more than 6 million people annually (*World Health Organization, 2011*). In the United States, the direct and indirect financial costs of smoking are estimated at more than $300 billion each year (*U.*

**eLife digest** Nicotine addiction is a serious public health problem. People who start using nicotine during adolescence are more likely to become addicted to it during adulthood, but the reasons for this are not well understood. Nicotine causes long-lasting changes in the brain that are responsible for the feelings of pleasure and reward. In particular, nicotine strengthens the connections between neurons at structures called synapses and increases communication between reward-related neurons in key reward areas of the brain. This hijacking of the natural reward system requires new proteins to be made. However, the relationship between protein synthesis and adolescents being particularly vulnerable to nicotine addiction was not known.

In a related study, Huang et al. found that the reduced activity of a protein called eIF2α, which controls the production of new proteins, accounts for why adolescents are more likely to become addicted to cocaine than adults. Thus, Placzek et al. wanted to know whether the same was true for nicotine and whether the proteins controlled by eIF2α are involved in the way human nicotine addicts experience reward.

Placzek et al. found that adolescent mice are more susceptible than adult mice to the changes in synaptic strength that are caused by nicotine. This increased susceptibility results from reduced activity levels of the protein eIF2α. Reducing the activity of eIF2α in adult mice made their synapses as likely to change strength in response to nicotine as the synapses of adolescent mice.

Placzek et al. also used a technique called functional magnetic resonance imaging and found that compared to non-smokers, the brain activity of human smokers was significantly reduced when given a natural reward. Further studies revealed a variation in the gene encoding the eIF2α protein that affects how smokers respond to a reward, suggesting that this variant is linked to the likelihood that a person will be addicted to nicotine.

This work raises several important questions. In addition to regulating the initial adaptive changes induced in the brain by nicotine, does eIF2α activity affect compulsive nicotine use? If so, could targeting parts of the eIF2α pathway help treat nicotine addiction? Finally, further studies could explore whether the gene variant identified by Placzek et al. affects how users of other drugs (such as cocaine or alcohol) respond to natural rewards.

S. Department of Health and Human Services, Centers for Disease Control and Prevention, National Center for Chronic Disease Prevention and Health Promotion, Office on Smoking and Health, 2014; Xu et al., 2015). Adolescents are particularly at risk for initiating tobacco use, with a vast majority of all smokers beginning at age 18 or younger (U.S. Department of Health and Human Services, Centers for Disease Control and Prevention, National Center for Chronic Disease Prevention and Health Promotion, Office on Smoking and Health, 2014). A growing body of evidence from both human and animal studies indicates that adolescents are more susceptible than adults to the cellular and behavioral effects of nicotine, the main addictive component of tobacco (Adriani and Laviola, 2004; Counotte et al., 2011; Lydon et al., 2014; Smith et al., 2015).

Nicotine modifies dopamine (DA) signaling in key regions of the brain's reward system (Jasinska et al., 2014; De Biasi and Dani, 2011). Human neuroimaging studies of smokers have shown that exposure to nicotine alters reward-related activity in dopaminergic reward regions (Rose et al., 2012). Moreover, in rodents, while nicotine is known to potentiate excitatory synaptic connections to DA neurons in the reward-related ventral tegmental area (VTA) (Mansvelder and McGehee, 2000; Saal et al., 2003), the precise molecular mechanism underlying nicotine-induced long-term potentiation (LTP) remains unclear. In a companion article (Huang et al., 2016), we found that the translation initiation factor eIF2α regulates adolescent vulnerability to the synaptic and behavioral effects of cocaine. Briefly, adolescent (5 weeks old) mice proved to be more sensitive to a lower dose of cocaine than adult (3–5 months old) mice with regard to reduced phosphorylation of eIF2α, the induction of LTP, and cocaine-induced behavior [conditioned place preference (CPP), a common behavioral task that reflects behavioral reinforcement underlying the development of drug addiction] (Koo et al., 2012). Consistent with these findings, genetic and pharmacological reduction

in p-eIF2α-mediated translational control increased the susceptibility of adult mice to the synaptic and behavioral effects of cocaine, making adult mice resemble adolescents in this respect. Furthermore, other drugs of abuse (including nicotine), that are known to induce LTP in the VTA (*Saal et al., 2003*), also reduced p-eIF2α in the VTA (*Huang et al., 2016*), thus highlighting the role of p-eIF2α as a common effector implicated in the initiation of addictive behavior. In the present study, we found that, like cocaine, mice with reduced p-eIF2α-mediated translation are more susceptible to nicotine-evoked synaptic potentiation in the VTA. Furthermore, using functional magnetic resonance imaging (fMRI) in humans, we identified a functional genetic variation in the promoter of the *Eif2s1* gene encoding eIF2α that alters brain responses to rewarding stimuli in human tobacco smokers.

## Results and discussion

### Adolescent mice are more susceptible than adult mice to nicotine-induced LTP in VTA DA neurons

In the accompanying study, we found that adolescent mice are more susceptible to cocaine-induced synaptic potentiation in VTA DA neurons. We therefore asked whether this was also true for nicotine. To answer this question, we measured glutamate-mediated excitatory postsynaptic currents (EPSCs) in VTA DA neurons from adolescent (5 weeks old) and adult (3–5 months old) mice 24 hr after single intra-peritoneal (i.p.) injection of either saline or different doses of nicotine. The peak amplitudes of the α-amino-3-hydroxy-5-methyl-4-isoxazolepropionic acid receptor (AMPAR) and *N*-methyl D-aspartate receptor (NMDAR)-mediated components of the evoked EPSCs (recorded at +40 mV) were isolated and used to calculate the AMPAR/NMDAR ratio as described (*Huang et al., 2016*). An increase in this ratio was taken as an index of LTP. We found that a relatively low dose of nicotine (0.4 mg/kg i.p.) was sufficient to induce LTP in VTA DA neurons from adolescent mice, but not in adult mice (*Figure 1a, b* and *Figure 1—figure supplement 1*). By contrast, a higher dose of nicotine (1.0 mg/kg) was required to elicit comparable LTP in VTA DA neurons from adult mice (*Figure 1b* and *Figure 1—figure supplement 1*). Thus, like cocaine, nicotine induces LTP in VTA DA neurons at a significantly lower dose in adolescent mice.

### Like adolescent mice, adult mice with reduced eIF2α-mediated translational control are more susceptible to nicotine-evoked LTP

In the accompanying article (*Huang et al., 2016*) we found that a low dose of cocaine reduces phosphorylation of eIF2α only in the VTA of adolescent mice. To test whether the same is true for nicotine, we injected adolescent and adult mice with a low (0.4 mg/kg) and a relatively high (1 mg/kg) dose of nicotine, respectively. Consistent with our findings with cocaine, we found that a low dose of nicotine (0.4 mg/kg) reduced p-eIF2α only in the VTA of adolescent mice (*Figure 1c, d*), whereas a higher dose of nicotine (1 mg/kg) was required to reduce p-eIF2α in the VTA of adult mice (*Figure 1e*). Thus, like cocaine, low doses of nicotine selectively reduce p-eIF2α in the VTA of adolescent mice, highlighting the involvement of p-eIF2α-mediated translational control during this period of heightened vulnerability to the effects of drugs of abuse.

Based on these findings, we predicted that adult mice with reduced p-eIF2α-mediated translational control would be more susceptible to the synaptic effects of nicotine. To test this prediction, we injected both adult control and *Eif2s1^{S/A}* mice (in which p-eIF2α in VTA is reduced by about 50% because the phosphorylation site is mutated to alanine (*Huang et al., 2016*) with a low dose of nicotine (0.4 mg/kg i.p.). This low dose does not typically induce LTP in adult wild-type (WT) mice (*Figure 1b*), and as expected, it failed to induce LTP in control WT *Eif2s1^{S/S}* mice (*Figure 1f*). By contrast, the same low dose of nicotine elicited LTP in adult *Eif2s1^{S/A}* mice (*Figure 1f*). Thus, like adolescent mice, adult mice with reduced eIF2α phosphorylation are more susceptible to the synaptic effects of nicotine.

To further support these findings, we used the recently discovered small molecule ISRIB (*Sidrauski et al., 2013*), which selectively blocks p-eIF2α-mediated translational control (*Sidrauski et al., 2013*). Briefly, adult WT mice were acutely injected with both ISRIB (2.5 mg/kg) and a low dose of nicotine (0.4 mg/kg) and LTP was recorded in VTA DA neurons 24 hr later. Indeed, a low dose of nicotine (0.4 mg/kg) induced LTP only in adult mice in which p-eIF2α-mediated

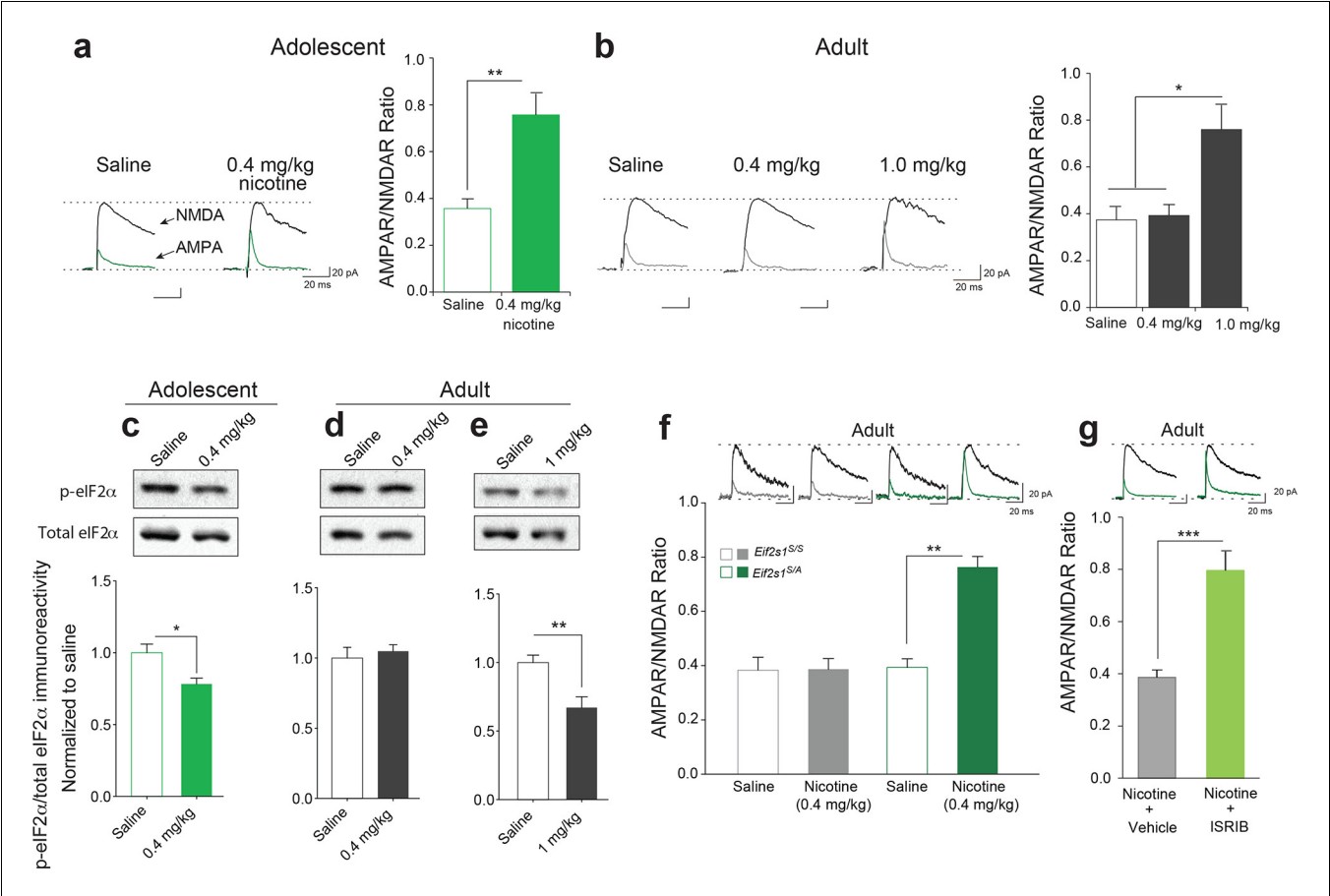

**Figure 1.** Reduced p-eIF2α-mediated translational control increases the susceptibility to nicotine-induced LTP. (**a-b**) Left, Representative traces of AMPAR and NMDAR EPSCs recorded from VTA DA neurons 24 hr after i.p. injection of saline or the indicated dose of nicotine. A relatively low dose of nicotine (0.4 mg/kg) induced LTP, shown by an increase in AMPAR/NMDAR ratio in VTA DA neurons (a, Right, $P<0.01$, $n=6,6$ saline/0.4 mg/kg nicotine, $t_{10}=4.026$) from adolescent mice (5 weeks old), but not in those from adult mice (3–5 months old, b, Right, $P=0.802$, $n=6/7/6$ saline/0.4 mg/kg nicotine/ 1.0 mg/kg nicotine, $F_{2,16}=9.029$). A higher dose of nicotine (1.0 mg/kg) was required to increase the AMPAR/NMDAR ratio in VTA DA neurons from adult mice (b, Right, $P<0.05$ vs. saline or 1.0 mg/kg nicotine, $n=6/7/6$ saline/0.4 mg/kg nicotine/1.0 mg/kg nicotine, $F_{2,16}=9.029$). (**c-d**) A low dose of nicotine (0.4 mg/kg) reduced p-eIF2α in the VTA of adolescents (c, $P<0.05$, $n=9/5$ saline/0.4 mg/kg nicotine, $t_{12}=2.479$), but not adult mice (d, $P=0.5710$, $n=7/11$ saline/0.4 mg/kg nicotine, $t_{16}=0.5784$). (**e**) A higher dose of nicotine (1 mg/kg) was required to reduce p-eIF2α in VTA of adult mice ($P<0.01$, $n=11/5$ saline/1 mg/kg nicotine, $t_{14}=3.428$). (**f**) A low dose of nicotine (0.4 mg/kg) failed to induce LTP in VTA DA neurons from adult WT (*Eif2s1*$^{S/S}$) mice (Left, $P=0.964$, $n=5$ per group, $t_8=0.05$), but elicited significant LTP in adult *Eif2s1*$^{S/A}$ mice (Right, $P=0.003$, $n=5$ per group, $t_8=6.73$). (**g**) A low dose of nicotine (0.4 mg/kg) induced LTP in ISRIB-injected adult mice compared to vehicle-injected mice ($P<0.001$, $n=7/7$ nicotine+vehicle/nicotine+ISRIB, $t_{12}=5.222$).

The following figure supplement is available for figure 1:

**Figure supplement 1.** Adolescent mice are more susceptible than adult mice to nicotine-induced synaptic potentiation.

translation was blocked pharmacologically with ISRIB (*Figure 1g*). Hence, like adolescent mice, adult mice with reduced p-eIF2α-mediated translational control are more susceptible to nicotine-induced LTP. Thus, our findings that reducing p-eIF2α-mediated translational control renders animals more susceptible to the effects of both cocaine (*Huang et al., 2016*) and nicotine underscores a key role of p-eIF2α as a common regulator of drug-induced synaptic potentiation and behavior.

## Polymorphic variation in the *Eif2s1* gene affects reward signaling measured by fMRI in human tobacco smokers

Several studies in rodents have shown that certain genes or signaling pathways are implicated in the behavioral effects of drugs of abuse (*Robison and Nestler, 2011*), but their clinical relevance to

humans remains unknown. Because p-eIF2α crucially regulates drug-induced changes in synaptic strength and behavior in mice, we sought to determine whether single nucleotide polymorphisms (SNPs) in the eIF2α signaling pathway could be associated with reward-induced changes in neuronal activity in human smokers. Indeed, by studying specific SNPs chosen on the basis of this hypothesis, we bypassed the problems inherent to large exploratory data analyses and multiple comparisons typical of genome-wide association studies. In this way we increased the chances of finding true effects related to biological processes, while reducing the possibility of false positives.

According to neuroimaging studies, neuronal activity in key reward areas of the human brain is strongly associated with indices of drug use (*Rose et al., 2014*). We therefore measured reward-mediated activity in the caudate and putamen—brain regions with crucial reward-related functional connections to the VTA (*Koob and Volkow, 2010*)—of tobacco smokers and non-smokers (*Figure 2—figure supplement 1*). To elicit reward responses in the fMRI scanner, participants received small (1 mL) squirts of sweet juice orally while functional MRI images of their brains were collected (see Material and Methods and *Figure 2—figure supplement 2*). Consistent with previous findings in cocaine and tobacco users (*Rose et al., 2012*; *Rose et al., 2014*), we observed significantly lower reward-induced activity in the caudate and putamen of smokers (*Figure 2a and c*), indicating that smokers find sweet drinks less rewarding. More importantly, we identified an SNP (rs10144417) in the *Eif2s1* (eIF2α) gene that revealed an interaction between genotype and smoking status. While smokers carrying the AG/GG genotype showed lower reward-dependent activity compared with that of the AA smokers, such activity did not differ between non-smokers of both genotypes (*Figure 2d*). These data indicate that rs10144417 in the *Eif2s1* gene is associated with both reward signaling and tobacco use.

In silico analysis revealed that rs10144417 spans a highly conserved region of the *Eif2s1* promoter (*Figure 2e*). To examine the functional effect of the A/G polymorphism (rs10144417) at this site, we measured firefly luciferase reporter activity ex-vivo using a 5 kilobase (kb) segment of the *Eif2s1* promoter (*Figure 2f*). Briefly, human embryonic kidney (HEK293T) cells were co-transfected with a firefly luciferase reporter (Fluc) containing either the A or G variant and a renilla luciferase reporter (Rluc), which was used as a transfection control. We found that, compared to the A variant, the normalized G variant Fluc/Rluc ratio was significantly increased (~40%, *Figure 2g*), reflecting increased expression of *Eif2s1*. To determine whether overexpression of *Eif2s1* could affect p-eIF2α–mediated translational control, we co-expressed in HEK293T cells *Eif2s1* (*Donzé et al., 1995*) (*Figure 2h*), a 5'UTR-*Ophn1* firefly luciferase reporter (5'UTR-*Ophn1*-Fluc; *Figure 2i*), whose translation is known to be enhanced by conditions that increase p-eIF2α (*Di Prisco et al., 2014*), and a renilla Luciferase reporter (Rluc), which was used to calculate the relative Fluc/Rluc translation ratio. Strikingly, increased expression of *Eif2s1* significantly reduced the normalized Fluc/Rluc ratio, indicating a significant reduction in translation of *Ophn1* mRNA (*Figure 2j*). Additionally, expression of *Eif2s1* selectively reduced 5'UTR-*Ophn1*-Fluc activity, but had no effect on Rluc activity (data not shown). These data were further supported by our findings that overexpression of *Eif2s1* also reduced translation of a 5'UTR-*Atf4* firefly luciferase reporter (*Figure 2—figure supplement 3*), which is typically up-regulated by increased p-eIF2α (*Sidrauski et al., 2013*; *Lu et al., 2004*). Hence, overexpression of the alpha subunit of eIF2 (*Eif2s1*) reduces p-eIF2α–mediated translation.

The mechanism by which increased expression of *Eif2s1* inhibits p-eIF2α–mediated translation remains to be determined. Several different mechanisms could be at play. Overexpression of *Eif2s1* could affect eIF2α-mediated translation by altering: a) the assembly of eIF2 complex [eIF2 is a heterotrimer consisting of an alpha (encoded by *Eif2s1*), a beta (encoded by *Eif2s2*), and a gamma subunit (encoded by *Eif2s3*), b) the binding of eIF2 to key regulatory proteins (e.g., eIF2B and/or eIF5), or c) by titrating the phosphorylated alpha subunit away from the eIF2 complex. Moreover, overexpression of *Eif2s1* could alter the expression of a given eIF2α kinase or phosphatase. Such compensatory translational homeostatic mechanisms have been observed when the levels of key translation initiation factors (e.g., eIF4E, 4E-BPs, PABP, Paips) are either increased or decreased (*Khaleghpour et al., 1999*; *Yanagiya et al., 2012*; *Yoshida et al., 2006*).

Collectively, our mouse and human data suggest that reduced p-eIF2α-mediated translational control mediates a genetic predisposition to greater risk for drug-induced changes in synaptic strength, which may account for the greater vulnerability of adolescents, even as first time drug users. For instance, mice with reduced eIF2α-mediated translation are more susceptible to nicotine-induced changes in synaptic function. Similarly, smokers carrying the G variant, who also have

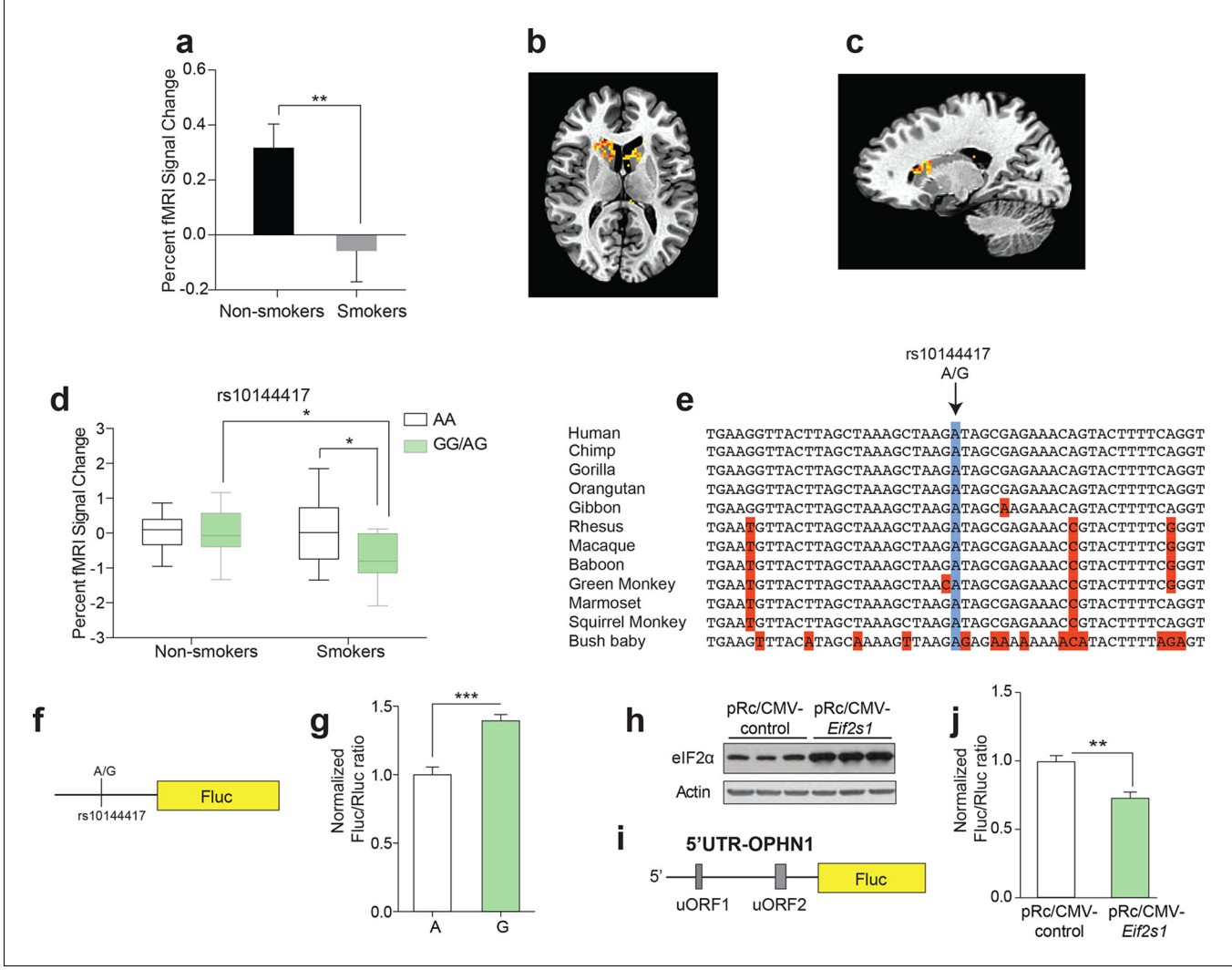

**Figure 2.** The effect of a single nucleotide polymorphism (SNP) in the promoter of the *Eif2s1* gene on reward-dependent striatal activity in human tobacco smokers. (**a-c**) Reward-related activity in caudate/putamen is lower in smokers than non-smokers (**b**, $P<0.01$, n=33/55, $t_{86}=2.678$). (**b-c**) Transverse (**b**) and sagittal (**c**) views of significant fMRI BOLD signal in caudate/putamen of non-smokers compared to smokers in response to juice reward. (**d**) Interaction between smoking and rs10144417 genotype ($P<0.05$, $F_{1,86}=5.836$). (**e**) Partial alignment of *Eif2s1* promoter sequences in human and related animals. Note high level of nucleotide conservation: the rs10144417 SNP is indicated in blue and the non-conserved nucleotides in red. (**f**) Schematic of firefly luciferase reporter constructs, in which a 5-kb *Eif2s1* promoter fragment containing either the A or G allele was cloned upstream of the firefly luciferase gene in the p15Amp reporter vector. A renilla luciferase reporter was co-transfected with reporters containing the A or G variant and firefly luciferase (Fluc) activity was normalized to renilla (Rluc) activity. (**g**) Effects of A and G variants of SNP rs10144417 on the transcriptional activity of the *Eif2s1* promoter as assessed by a luciferase reporter assay in HEK-293 cells. The data are from three independent experiments ($P<0.001$, n=6 per group, $t_{10}=5.405$). (**h**) Western blotting showing that overexpression of *Eif2s1* (pRc/CMV-*Eif2s1*) increased eIF2α levels compared to control (vector alone pRc/CMV). (**i**) Diagram of the 5′ UTR-*Ophn1*-Fluc reporter, which consists of the 5′UTR of *Ophn1* mRNA fused to the coding region of Firefly luciferase (Fluc). A renilla luciferase (Rluc) reporter vector was co-transfected into HEK293T as a transfection control. (**j**) Overexpression of *Eif2s1* reduced expression of 5′ UTR-*Ophn1*-Fluc ($P<0.01$, n=6, $t_{10}=3.9425$).

The following figure supplements are available for figure 2:

**Figure supplement 1.** Demographic information of human participants involved in fMRI studies.

**Figure supplement 2.** fMRI recording paradigm and reward-stimulus pairing in human smokers.

**Figure supplement 3.** ATF4-Luciferase construct design and activity with *Eif2s1*.

reduced eIF2α-mediated translation, show reduced reward-induced activity in the caudate and putamen, suggesting that such individuals are likely to consume more drugs to obtain reward activity comparable to that of non-smokers. Finally, these insights may hold promise for new p-eIF2α-based approaches to treating drug abuse.

## Materials and methods

### Mice

All experiments were conducted on male and female mice from the C57BL/6 background. *Eif2s1$^{S/A}$* mice were previously described (*Di Prisco et al., 2014*). Mice were kept on a 12h/12 hr light/dark cycle (lights on at 7:00 am) and had access to food and water *ad libitum*. Animal care and experimental procedures were approved by the institutional animal care and use committee (IACUC) at Baylor College of Medicine, according to NIH Guidelines.

No statistical methods were used to predetermine sample sizes. All sample sizes met the criteria for corresponding statistical tests—our sample sizes are similar to those reported in previous publications (*Bellone and Lüscher, 2006*; *Ungless et al., 2001*).

### Drug treatment

Nicotine was dissolved in 0.9% saline and injected at a volume of 5 ml/kg. (-)-Nicotine hydrogen tartrate was obtained from Sigma-Aldrich (St. Louis, MO). ISRIB (provided by P. Walter) was dissolved in DMSO and further diluted in PEG-400 (1:1 ratio) as previously described (*Sidrauski et al., 2013*). For both electrophysiological and behavioral experiments, ISRIB (2.5 mg/kg) or vehicle (DMSO/PEG-400, 2 ml/kg) was injected 90 min before nicotine or control saline injections.

### Slice electrophysiology

Electrophysiological recordings were performed as previously described (*Ungless et al., 2001*), investigators blind to genotype. Each electrophysiological experiment was replicated at least three times. Briefly, mice were anesthetized with a mixture of ketamine (100 mg/kg), xylazine (10 mg/kg), and acepromazine (3 mg/kg). Mice were transcardially perfused with an ice-cold, oxygenated solution containing (in mM) NaCl, 120; NaHCO$_3$, 25; KCl, 3.3; NaH$_2$PO$_4$, 1.2; MgCl$_2$, 4; CaCl$_2$, 1; dextrose, 10; sucrose, 20. Horizontal slices (225–300 µm thick) containing the VTA were cut from the brains of adolescent (5 week old) or adult (3–5 month old) C57BL/6J mice using a vibrating tissue slicer (VF-100 Compresstome, Precisionary Instruments, San Jose, CA, or Leica VT 1000S, Leica Microsystems, Buffalo Grove, IL). Slices were next incubated at 34°C for 40 min then kept at room temperature for at least 30 min before they were transferred to a recording chamber where they were continuously perfused with artificial cerebrospinal fluid (ACSF) at 32°C at a flow rate of 2–3 ml/min. The recording ACSF differed from the cutting solution in regard to the concentration of MgCl$_2$ (1 mM) and CaCl$_2$ (2 mM). Recording pipettes were made from thin-walled borosilicate glass (TW150F-4, WPI, Sarasota, FL). After filling with intracellular solution (in mM): 117 CsMeSO3; 0.4 EGTA; 20 HEPES; 2.8 NaCl, 2.5 ATP-Mg 2.0; 0.25 GTP-Na; 5 TEA-Cl, adjusted to pH 7.3 with CsOH and 290 mOsmol/l, they had a resistance of 3–5 MΩ.

Data were obtained with a MultiClamp 700B amplifier, digitized at 20 kHz with a Digidata 1440A, recorded by Clampex 10 and analyzed with Clampfit 10 software (Molecular Devices). The signals were filtered online at 4 kHz with a Bessel low-pass filter. A 2 mV hyperpolarizing pulse was applied before each EPSC to evaluate the input and access resistance (Ra). Data were discarded when Ra was either unstable or greater than 25 MΩ, holding current was > 200 pA, input resistance dropped > 20% during the recording, or baseline EPSCs changed by > 10%. Traces illustrated in Figures are averages of 10–15 consecutive traces.

After establishing a gigaohm seal (> 2 GΩ) and recording stable spontaneous firing in cell-attached voltage clamp mode (at -70 mV holding potential), cell phenotype was determined by measuring the width of the action potential (*Ford et al., 2006*). AMPAR/NMDAR ratios were calculated as previously described (*Ungless et al., 2001*). Briefly, neurons were voltage-clamped at +40 mV until the holding current stabilized (at < 200 pA). Monosynaptic EPSCs were evoked at 0.05 Hz with a bipolar stimulating electrode placed 50–150 µm rostral to the lateral VTA. Picrotoxin (100 µM) was added to the recording ACSF to block GABA$_A$R-mediated IPSCs. After recording the dual-

component EPSC, DL-AP5 (100 μM) was bath-applied for 10 min to remove the NMDAR component, which was then obtained by offline subtraction of the remaining AMPAR component from the original EPSC. The peak amplitudes of the isolated components were used to calculate the AMPAR/NMDAR ratios. Picrotoxin and DL-AP5 were purchased from Tocris Bioscience and all other reagents and experimental compounds were obtained from Sigma-Aldrich.

## Plasmid construction, transfection and luciferase assay

The *Eif2s1* promoter region was cloned out of human BAC RP11-713C11 using primers 5' sense 5'-ATTCGCGAGGGAAAGATTTCAATTC-3', and antisense 5'-TCTGCAATTTAAACAAAAGAATTAAG-TAAGT-3'. The firefly luciferase was amplified from Luciferase-pcDNA3 (Promega, Madison, WI) using the sense primer 5'-ACTTACTTAATTCTTTTGTTTAAATTGCAGAATGGAAGACGCCAAAAAC-ATAAAG3' and antisense primer 5'-ACATTTCCCCGAAAAGTGCCACCTGCCATAGAGCCCACCGC-ATCCCCAG-3'. The p15a_Amp was amplified from pWSTK6 using sense primer 5'-CAGGTGGCACTTTTCGGGGAAATGT-3' and antisense primer 5'-GAATTGAAATCTTTCCCTCGCG-AATGCTAGCGGAGTGTATACTGGCTTAC-3'. All three fragments were Gibson cloned and confirmed by DNA sequencing. The resulting plasmid (*Eif2s1*-A-luc) carries the A in the SNP rs101444417 position up-stream of firefly luciferase. In a mutagenesis reaction using the *Eif2s1*-A-luc plasmid as template and sense primer 5'-GGTTACTTAGCTAAAGCTAAGGTAGCGAGAAACAGTA-CTTTTCAG-3' and antisense primer 5'-CCTGAAAAGTACTGTTTCTCGCTACCTTAGCTTTAGCTAA-GTAACC-3', we generated a clone (*Eif2s1*-G-luc) containing the G in the SNP rs101444417 position up-stream of firefly luciferase. HEK293T cells (a widely used human cell line) were grown in 24-well plates in Gibco DMEM+Glutamax (Life Technologies) supplemented with 10% FBS, 100 units of Pen/Strep per ml. *Eif2s1*-G-luc and *Eif2s1*-A-luc were co-transfected with a renilla luciferase (Rluc) plasmid pRL-TK (Promega, Madison, WI) into HEK293T cells at 50–80% confluency using Lipofectomine LTX plus (Life Technologies). pRc/CMV and pRc/CMV-*eIF2s1* vectors, which were previously reported (*Donzé et al., 1995*), were co-transfected with 5'UTR-*Ophn1*-Fluc and RLuc into HEK293T, as described above. Similarly 5'UTR-*Atf4*-Fluc (*Sidrauski et al., 2013*) was co-transfected with RLuc into HEK293T cells. 24 hr after transfection, cell extracts were prepared in passive lysis buffer and samples were collected in pre-chilled microcentrifuge tubes and lysed in homogenizing buffer [200 mM HEPES, 50 mM NaCl, 10% Glycerol, 1% Triton X-100, 1 mM EDTA, 50 mM NaF, 2 mM $Na_3VO_4$, 25 mM β-glycerophosphate, and EDTA-free complete ULTRA tablets (Roche, Indianapolis, IN)].

## Human functional MRI data

All procedures were approved by Baylor College of Medicine Internal Review Board. Smokers and non-smokers were recruited from the Houston metropolitan area via fliers, newspaper, and internet advertisements and received a small monetary compensation. All participants were pre-screened to rule out non-tobacco substance dependence or MRI contraindications (e.g. head injuries, foreign metal in the body, claustrophobia, or pregnancy). Smokers currently seeking cessation treatment were also excluded. Smoking history and dependence were evaluated using the Fagerström Test for Nicotine Dependence (FTND) (*Heatherton et al., 1991*), Shiffman-Jarvik Withdrawal Questionnaire (SJWQ) (*Shiffman and Jarvik, 1976*), and the Positive and Negative Affect Schedule (PANAS) (*Watson and Clark, 1994*). All participants read and signed an Informed Consent to participate in this research protocol.

No statistical methods were used to predetermine sample sizes but sample sizes were similar to those previously reported (*Rose et al., 2012*; *Salas et al., 2010*). The subjects were either smokers (*n*=35, average age=43.26 years ± 12.19 years, 77.1% males, 22.9% females, smoking at least 10–15 cigarettes per day and had smoked for at least the past year) or non-smokers (*n*=54, average age=31.37 ± 11.60 years, 44.4% males 55.5% females, lifetime incidence of smoking less than 50 cigarettes or no cigarettes for the last year). For more information on participant demographic, see *Figure 2—figure supplement 1*.

Participants were scanned in 3T Siemens Trio scanners in the Center for Advanced MRI at Baylor College of Medicine. Structural MPRAGE images were collected as 160 1×1×1 mm axial slices (TE=2.66 ms, TR=1200 ms, flip angle=12°, 256 × 256 matrix) while functional images were collected as 2x2x2 mm epi scans (TR 2 s, TE 40 ms) over a 44 mm "slab" covering the regions of interest. Light-juice pairings were presented during functional scanning using ePrime software (Psychology

Software Tools, Sharpsburg, PA). For each of a maximum of 55 light-juice pairings, a Standard Infuse/Withdraw Harvard 33 Twin Syringe Pump (Harvard Apparatus, Holliston, MA) delivered 1 mL of juice 7s after a light cue. Prior to scanning, participants were given a choice of sugar-free sweet drinks (such as lemonade, iced tea with peach or fruit punch, and the preferred juice was given during scanning). fMRI data were analyzed using a standard AFNI processing stream (*Cox, 1996*). Briefly, the first four TRs were removed to establish a stable baseline. Data were then corrected for slice-time acquisition (3dTshift), aligned to the first image and measured for motion (3dvolreg), registered to the high resolution MPRAGE, and transformed to MNI space using a single spatial transform (@auto_tlrc, 3dAllineate). A 4.5 mm smoothing kernel was applied in 3dmerge and submitted to a General Linear Model (GLM) regression in 3dDeconvolve. GLM regressors for linear, quadratic, and cubic linear trends; x, y, z, roll, pitch, and yaw motion parameters; and the two stimulus conditions: visual cue and juice reward were included. An analysis of the interaction between smoker/non-smoker group and genotype (e.g. rs10144417 AA vs. GG/AG) was performed using a 3D multivariate model (3dMVM) and family-wise error correction. The percent signal change in BOLD activation for individual subjects was extracted from caudate and putamen regions of interest (ROIs) and restricted to those voxels identified as having a $P<0.05$ and alpha$<0.05$ in the multivariate model of group by genotype interaction. No effect of age, number of years smoking, average number of cigarettes per day, gender, or ethnicity was observed in the fMRI analysis.

## Human genotype

DNA was isolated from buccal swabs and genotyping was completed using Illumina (San Diego, CA) HumanOmniExpress-12 v1.1 BeadChip arrays (cat# WG-312-1120) containing approximately 741K SNPs. Genomic DNA (200 ng) from each sample was processed following Illumina's Infinium HD Ultra Assay protocol. BeadChip images were captured using Illumina iScan System. The Illumina chip contained the *Eif2s1* rs10144417 SNP (Global minor allele frequency MAF:0.2993). fMRI activity was studied according to the genotype of analyzed subjects. Subjects with AG and GG genotypes were pooled for analysis and compared to subjects with the AA genotype using AFNI's statistical software.

## Statistical analyses

All data are presented as mean ± s.e.m. Statistical analyses were performed with SigmaPlot (Systat Software). Data distribution normality and homogeneity of variance were assessed by Shapiro-Wilk and Levene tests, respectively. The statistics were based on the two-sided Student's t test, or one- or two-way ANOVA with Tukey's HSD (or HSD for unequal sample sizes where appropriate) to correct for multiple *post hoc* comparisons. Within-groups variation is indicated by standard errors of the mean of each distribution, which are depicted in the graphs as error bars. $P<0.05$ was considered significant (*$P<0.05$, **$P<0.01$, ***$P<0.001$, ****$P<0.0001$).

## Acknowledgements

We thank Hongyi Zhou for assisting in the maintenance of the mouse colony and members of the Costa-Mattioli laboratory for comments on the manuscript. This work was supported by grants from the National Institutes of Health to MCM (NIMH 096816, NINDS 076708), JD (NIDA DA09411 and NINDS NS21229) and RS (NIDA026539 and NIDA09167) and from the VA to RS (VHA5I01CX000994). PW is an Investigator of the Howard Hughes Medical Institute.

## Additional information

### Funding

| Funder | Grant reference number | Author |
| --- | --- | --- |
| National Institute of Mental Health | MH096816 | Mauro Costa-Mattioli |
| National Institute of Neurological Disorders and Stroke | NS076708, NS21229 | John A Dani<br>Mauro Costa-Mattioli |

| National Institute on Drug Abuse | DA09411, DA026539, DA09167 | John A Dani<br>Ramiro Salas |
| Howard Hughes Medical Institute | PW | Peter Walter |
| U.S. Department of Veterans Affairs | VHA5I01CX000994 | Ramiro Salas |

The funders had no role in study design, data collection and interpretation, or the decision to submit the work for publication.

## Author contributions

ANP, Designed and performed electrophysiological experiments and wrote the manuscript, Conception and design, Acquisition of data, Analysis and interpretation of data, Drafting or revising the article; DLM, Designed and performed fMRI experiments, Conception and design, Analysis and interpretation of data; SK, Designed and performed luciferase experiments and immunoblotting experiments, Conception and design, Acquisition of data; GVDP, Performed electrophysiological experiments, Acquisition of data; WH, Design initial experiments on drugs of abuse and p-eIF2$\alpha$ phosphorylation, Conception and design; CS, Contributed to discussion of the project, Drafting or revising the article; KK, PW, Contributed to in-depth discussion of the project and editing of the manuscript, Drafting or revising the article; CLA, Performed human genotyping, Acquisition of data; RR, Helped with the luciferase experiments, Acquisition of data; JAD, Contributed to in-depth discussion of the project, Conception and design; RS, Analyzed the fMRI data and wrote the manuscript, Analysis and interpretation of data, Drafting or revising the article; MCM, Designed experiments and wrote the manuscript, Conception and design, Analysis and interpretation of data, Drafting or revising the article

## Author ORCIDs

Mauro Costa-Mattioli, (iD) http://orcid.org/0000-0002-9809-4732

## Ethics

Human subjects: All procedures were approved by Baylor College of Medicine Internal Review Board (IRB, H-27725). Participants were recruited for the study and received a small monetary compensation. All participants read and signed an Informed Consent to participate in this research protocol.
Animal experimentation: Animal care and experimental procedures were approved by the institutional animal care and use committee (IACUC) at Baylor College of Medicine (IACUC, AN-5068) according to NIH Guidelines.

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
