## [Decision Letter]

Thank you for submitting your work entitled "eIF2α regulates nicotine-evoked synaptic potentiation in mice and reward-induced neuronal responses in human smokers" for consideration by *eLife*. Your article has been reviewed by two peer reviewers, and the evaluation has been overseen by a Reviewing Editor and a Senior Editor.

The reviewers have discussed the reviews with one another and the Reviewing Editor has drafted this decision to help you prepare a revised submission.

1) The authors should be clearer about what their experiments show and do not show. For example, they do not have any evidence, as they claim, that "reduced p-eIF2α-mediated translational control is a genetic modifier of risk for drug addiction". The study did not seem to look at drug addiction in either mice or human subjects. Instead the authors studied reward responses. There is an important and clear distinction between the two.

2) Regarding the results from rodents, the authors should provide a direct evidence for the phosphorylation level of eIF2alpha in adolescent mice versus adult ones, as well as after nicotine treatment. It is arguable that changes in eIF2 α activity can underline the differential effect nicotine-mediated in adolescent mice compared to adults. A direct assay exploring this issue, if it can be performed rapidly, could be of relevance to further support the electrophysiological finding.

3) Did the authors test the higher dose of nicotine in adolescent mice? It would be an additional and interesting information whether in adolescent mice the nicotine effect has already reached a saturated level or the LTP could further increase at higher concentration of nicotine.

4) Although the nicotine-mediated effect on AMPAR/NMDAR ratio is clearly significant, the number of 4 observations is too low and only slightly above the minimal required number for the statistical power. Strong recommendation to increase the N.

5) The selection criteria of participants, especially smoker subjects, should be specified. Did they have history of any other drug dependence? A table depicting the participants' demographics would be useful for readers.

6) The presentation of the polymorphism data was confusing. Especially the relationship between the mouse work and human results should be made clear and caveats in the link pointed out. It is important to note that the strength of the paper is the connection between the mouse and human data and anything that would strengthen that connection would consequently strengthen the paper. The relation between FLUC and RLUC assays is also murky and confusing. This should also be explained better. Finally, the reviewers wonder which mechanisms could cause increased expression of Eif2s inhibits p-eIF2α−mediated translation.

---

## [Author Response]

*1) The authors should be clearer about what their experiments show and do not show. For example, they do not have any evidence, as they claim, that "reduced p-eIF2α-mediated translational control is a genetic modifier of risk for drug addiction". The study did not seem to look at drug addiction in either mice or human subjects. Instead the authors studied reward responses. There is an important and clear distinction between the two.*

We agree with the reviewers and have now replaced “drug addiction” by “drug-induced synaptic potentiation and behavior”. The sentence quoted above now reads as follows: “Collectively, our mouse and human data suggest that reduced p- eIF2α-mediated translational control mediates a genetic predisposition to greater risk for drug-induced changes in synaptic strength, which may account for the greater vulnerability of adolescents, even as first time drug users” (Results and Discussion).

*2) Regarding the results from rodents, the authors should provide a direct evidence for the phosphorylation level of eIF2alpha in adolescent mice versus adult ones, as well as after nicotine treatment. It is arguable that changes in eIF2α activity can underline the differential effect nicotine-mediated in adolescent mice compared to adults. A direct assay exploring this issue, if it can be performed rapidly, could be of relevance to further support the electrophysiological finding.*

As recommended by the reviewers, we administered different doses of nicotine to adult and adolescent mice and found that a low dose of nicotine reduces eIF2α phosphorylation in the VTA of adolescents, but not adult mice (see new Figure 1 and subsection “Like adolescent mice, adult mice with reduced eIF2α-mediated translational control are more susceptible to nicotine-evoked LTP”, first paragraph). Thus, like cocaine, our new experiments with nicotine highlight the involvement of p-eIF2α phosphorylation-mediated translational control during this period of heightened vulnerability to the effects of drugs of abuse.

While basal p-eIF2α levels are similar in the VTA of adults and adolescent mice (see new Figure 1—figure supplement 4 in the accompanying paper by Huang et al.), phosphorylation of eIF2α in the brain is dynamically regulated by kinases: double-stranded RNA-dependent kinase (PKR), general control non-derepressible 2 (GCN2), PKR-like endoplasmic reticulum kinase (PERK) and Heme-regulated eIF2α kinase (HRI); as well as by two phosphatase complexes: protein phosphatase 1/growth arrest and DNA-damage inducible protein (PP1/GADD34) and PP1/constitutive reverter of eIF2α phosphorylation (PP1/CReP)_1,2_. Thus, in the Discussion section of our accompanying paper (p. 12), we speculate that nicotine (and other drugs of abuse) either inhibit the activity of a given eIF2α kinase or promote the activity of eIF2α phosphatases, processes to which we suspect adolescents are more prone. We are currently investigating the precise mechanism by which drugs of abuse decrease p-eIF2α in the VTA and other reward-related areas.

*3) Did the authors test the higher dose of nicotine in adolescent mice? It would be an additional and interesting information whether in adolescent mice the nicotine effect has already reached a saturated level or the LTP could further increase at higher concentration of nicotine.*

As now shown in Figure 1—figure supplement 1, similar LTP is produced in adolescent mice by both a low (0.4 mg/kg) and a high dose of nicotine (1 mg/kg), indicating that even at lower doses of nicotine the LTP reaches a plateau in adolescence.

*4) Although the nicotine-mediated effect on AMPAR/NMDAR ratio is clearly significant, the number of 4 observations is too low and only slightly above the minimal required number for the statistical power. Strong recommendation to increase the N.*

We agree with the reviewers and have increased the N. Even with a larger sample size, the conclusion of our work remained unchanged.

*5) The selection criteria of participants, especially smoker subjects, should be specified. Did they have history of any other drug dependence? A table depicting the participants' demographics would be useful for readers.*

As suggested by the reviewers, we have now provided a supplemental figure that includes two tables with relevant information about our subjects (demographic, gender and age of the individuals; see Figure 2—figure supplement 1).

6) The presentation of the polymorphism data was confusing. Especially the relationship between the mouse work and human results should be made clear and caveats in the link pointed out. It is important to note that the strength of the paper is the connection between the mouse and human data and anything that would strengthen that connection would consequently strengthen the paper. The relation between FLUC and RLUC assays is also murky and confusing. This should also be explained better. Finally, the reviewers wonder which mechanisms could cause increased expression of Eif2s inhibits p-eIF2α−mediated translation.

As suggested by the reviewers, we have better explained the connection between the human and mouse data (subsection “Polymorphic variation in the Eif2s1 gene affects reward signaling measured by fMRI in human tobacco smokers”, last paragraph). In addition, we have made clearer the description of the luciferase experiments (in the third paragraph of the aforementioned subsection).

Finally, in the Discussion section, we now hypothesize different mechanisms by which expression of Eif2s could block eIF2α−mediated translation (subsection “Polymorphic variation in the Eif2s1 gene affects reward signaling measured by fMRI in human tobacco smokers”). As this is beyond the scope of the current paper, our intention is to further investigate this process in detail and publish the results separately.

References

1) Buffington, S. A., Huang, W. & Costa-Mattioli, M. Translational control in synaptic plasticity and cognitive dysfunction. Annual Review of Neuroscience37, 17-38, doi:10.1146/annurev-neuro- 071013-014100 (2014).

2) Dever, T. E., Dar, A. C. & Sicheri, F. in Translational Control in Biology and Medicine Cold Spring Harbor monograph series*, 39* (eds M. B. Mathews, N. Sonenberg, & J. W. B. Hershey) 319-345 (Cold Spring Harbor Laboratory Press, 2007).